# Three to Tango: Inhibitory Effect of Quercetin and Apigenin on Acetylcholinesterase, Amyloid-β Aggregation and Acetylcholinesterase-Amyloid Interaction

**DOI:** 10.3390/pharmaceutics14112342

**Published:** 2022-10-30

**Authors:** Irene Álvarez-Berbel, Alba Espargaró, Antonio Viayna, Ana Belén Caballero, Maria Antònia Busquets, Patrick Gámez, Francisco Javier Luque, Raimon Sabaté

**Affiliations:** 1Department of Pharmacy and Pharmaceutical Technology and Physical-Chemistry, School of Pharmacy and Institute of Nanoscience and Nanotechnology (IN2UB), University of Barcelona, 08028 Barcelona, Spain; 2Department of Nutrition, Food Sciences and Gastronomy, School of Pharmacy, Institute of Theoretical and Computational Chemistry (IQTCUB) and Institute of Biomedicine (IBUB), Campus Torribera, University of Barcelona, Prat de la Riba 171, 08921 Santa Coloma de Gramenet, Spain; 3Department of Inorganic and Organic Chemistry, Faculty of Chemistry, Institute of Nanoscience and Nanotechnology (IN2UB) and NanoBIC, University of Barcelona, 08028 Barcelona, Spain; 4Catalan Institution for Research and Advanced Studies, Passeig Lluís Companys 23, 08010 Barcelona, Spain

**Keywords:** antiamyloid, anti-Alzheimer, apigenin, quercetin, hydroxyflavones, polyphenols, amyloid, molecular dynamics, MM-GBSA

## Abstract

One of the pathological hallmarks of Alzheimer’s disease (AD) is the formation of amyloid-β plaques. Since acetylcholinesterase (AChE) promotes the formation of such plaques, the inhibition of this enzyme could slow down the progression of amyloid-β aggregation, hence being complementary to the palliative treatment of cholinergic decline. Antiaggregation assays performed for apigenin and quercetin, which are polyphenolic compounds that exhibit inhibitory properties against the formation of amyloid plaques, reveal distinct inhibitory effects of these compounds on Aβ40 aggregation in the presence and absence of AChE. Furthermore, the analysis of the amyloid fibers formed in the presence of these flavonoids suggests that the Aβ40 aggregates present different quaternary structures, viz., smaller molecular assemblies are generated. In agreement with a noncompetitive inhibition of AChE, molecular modeling studies indicate that these effects may be due to the binding of apigenin and quercetin at the peripheral binding site of AChE. Since apigenin and quercetin can also reduce the generation of reactive oxygen species, the data achieved suggest that multitarget catechol-type compounds may be used for the simultaneous treatment of various biological hallmarks of AD.

## 1. Introduction

Alzheimer’s disease (AD) is ranked as the third cause of death in the world after heart diseases and cancer, and is the most common cause of adult-onset dementia, affecting currently around 50 million people worldwide [1]. This neurodegenerative disorder is characterized by a gradual decline of cognitive functions, such as the loss of memory and learning skills, eventually disabling daily life activities and intellectual functions, in conjunction with depression and aggressive behavior [1,2,3].

The multifactorial nature of AD is reflected by the distinct hypotheses that have been proposed for its etiology, e.g., the cholinergic neuron damage, the amyloid-β (Aβ) cascade, oxidative stress and inflammation, among others [4]. The cholinergic hypothesis relates acetylcholine (ACh) deficiency to the deterioration of learning and memory [1,4,5]. Cholinesterase inhibitors (ChE-Is) have therefore been investigated as potential agents to prevent neurotransmission dysregulation [6,7,8]. In this context, donepezil, rivastigmine and galantamine are ChE-Is currently approved for the symptomatic treatment of AD [9,10]. On the other hand, the proteolytic cleavage of the amyloid precursor protein (APP) generates amyloid-β (Aβ) peptides characterized by variable sequences of 36–43 amino acids [11]. The length of the cleaved fragments affects both cell toxicity and aggregation, the most fibrillogenic 42 amino-acid-long peptide (Aβ42) being mainly found in the deposits observed in AD patients. The aberrant self-assembly of Aβ peptides leads to nonsoluble, off-pathway fibrillar aggregates through a multistep mechanism that comprises (primary/secondary) nucleation, condensation (or elongation) and polymerization (or fibrillation) [12]. Aβ40 and Aβ42 are the main components of the senile plaques. Since antiamyloid drugs tested in vitro show very similar effects for both peptides, the physicochemical properties of Aβ40 (viz. higher solubility and lower aggregation propensity relative to Aβ42) make this peptide to be a suitable choice for exploring the aggregation kinetics and the effect of potential antiaggregating agents by means of in vitro assays [13,14]. Interestingly, it has been noticed that acetylcholinesterase (AChE) consistently colocalizes with the amyloid deposits and accelerates the Aβ peptide assembly into Alzheimer-type aggregates, increasing their neurotoxicity [15,16]. Furthermore, aggregation experiments carried out in the presence of AChE inhibitors (AChEIs) directed against the catalytic (CAS) and peripheral (PAS) binding sites of AChE showed that only the PAS inhibitors are able to block the effect of AChE on amyloid formation [17,18,19].

As a result of the complex etiology of AD, intense research is focused on the development of multitarget compounds that encode the linkage of both synthetic and naturally occurring scaffolds [20,21,22]. Among natural compounds, flavonoids have been investigated due to their known antioxidant molecules and wide distribution in fruits and vegetables, making them common components of the human diet [23,24]. Several preclinical studies have highlighted their potential benefits in the treatment of AD [17,23,24]. In addition to the antioxidant activity, these benefits have been attributed to their activity as inhibitors of ChE [25,26], as well as to their Aβ antiaggregating effect [27,28]. AChE inhibitory activities in the micromolar range have been reported, and it has been shown that the IC50 values were dependent on the substitution pattern of the chemical scaffold [26,29]. Nevertheless, less attention has been paid to the relationship between the inhibitory activities and the molecular structure of these compounds. On the other hand, certain flavonoids like (-)-epigallocatechin-3-gallate are able to disrupt Aβ fibrillization, generating off-target oligomers [30]. Remarkably, several studies have pointed out that the presence of a catechol moiety might be responsible for the formation of covalent adducts with Aβ [27,31,32]. However, more studies are needed to elucidate the specific mechanisms of action of such flavonoids that prevent the formation of Aβ aggregates.

In this context, the present study aims at providing a comprehensive analysis of the potential AChE inhibition and Aβ antiaggregating activity of flavonoids, paying particular attention to the potential effect of AChE in promoting the formation of Aβ fibrils and how the presence of flavonoids may affect the interaction between AChE and Aβ [15,16]. To this end, apigenin and quercetin (Figure 1) have been chosen as representative flavonoids, and their activity against AChE and Aβ aggregation has been determined by using a combination of in vitro/in cellulo and in silico studies. The analysis is based on (i) the study of the role played by these molecules on the aggregation of Aβ40, through the determination of the time-course kinetics parameters, and the analysis of the molecular aggregates obtained by dynamic light scattering, infrared spectroscopy and transmission electron microscopy; (ii) the investigation of their effect on the AChE-induced aggregation of Aβ; (iii) the analysis of the inhibition of the AChE activity, considering the inhibition mechanism of the enzymatic activity and the ligand-induced quenching of the enzyme fluorescence; and (iv) finally, the study of the antioxidant activity (i.e., analysis of the production of free radicals in their presence).

## 2. Materials and Methods

### 2.1. Reagents

Aβ40 was purchased from Bachem (Bubendorf, Switzerland). Thioflavin-T (ThT), acetylcholinesterase from *Electrophorus electricus* (AChE), acetylthiocholine iodide (>99.0%), 5,5′-dithiobis(2-nitrobenzoic acid) (DTNB, >98.0%), quercetin and apigenin were obtained from Sigma-Aldrich. All solutions were prepared using native buffer (50 mM TRIS; 150 mM NaCl; pH 7.5).

### 2.2. Preparation of Aggregate-Free Aβ

Aβ40 (1 mg) was solubilized in 500 μL of 1,1,1,3,3,3-hexafluoro-2-propanol (HFIP) under vigorous stirring at room temperature for 1 h. The resulting solution was sonicated for 30 min and subsequently stirred at room temperature for 1 h. The solution was maintained at 4 °C for 30 min to avoid solvent evaporation during aliquot collection. To eliminate possible insoluble materials, the samples were filtered over 0.22 μm filters. Lastly, aliquots of soluble Aβ were collected, and the HFIP was evaporated under a gentle stream of dinitrogen. The samples were stored at −20 °C.

### 2.3. In Vitro Aβ40 Aggregation Assays

Aβ samples were resuspended in 50 μL of DMSO, and the monomers were solubilized through sonication for 10 min. Typically, a solution containing Aβ40 (20 µM), 50 µL of DMSO, 850 µL of native buffer and 100 µL of ThT is used to study the formation of fibrils. This solution is split into two parts; 10 µL AChE (from a stock solution of 200 μM) is added to one of them. These two solutions (V = 500 µL) are split again giving four solutions (V = 250 µL), two of them containing AChE. A 0.5 µL volume of inhibitor (10 mM) was added to one of the samples containing AChE. The samples are subsequently incubated at 37 °C for 24 h. The controls are incubated at 4 °C for 24 h. Fluorescence spectroscopy with ThT (λ_ex_ = 445 nm; λ_em_ = 480 nm) was used to detect the formation of Aβ fibrils with an Aminco-Bowman Series 2 luminescence spectrophotometer (Aminco-Bowman AB2, SLM Aminco, Rochester, NY, USA).

### 2.4. Aggregation Assay Analysis

The amyloid aggregation is analyzed considering an autocatalytic reaction, applying Equation (1):(1)$$f=\frac{\rho \left\{exp\left[\left(1+\rho \right)kt\right]-1\right\}}{1+\rho \cdot exp\left[\left(1+\rho \right)kt\right]}$$
where *f* is the fraction of Aβ in fibrillar form; the rate constant *k* includes the kinetic contributions arising from the formation of nuclei from monomeric Aβ, and the elongation of fibrils, which are, respectively, described by the rate constants *k*_n_ and *k*_e_; and *ρ* is a dimensionless parameter that describes the *k*_n_/*k* ratio. Equation (1) is obtained under the boundary conditions of *t* = 0 and *f* = 0, where *k* = *k*_e_*a* (*a* being the protein concentration). By nonlinear regression of *f* against *t*, the values of *ρ* and *k* can be determined, and from them, *k*_e_ and *k*_n_, which are the elongation and nucleation kinetic rate constants, respectively [33]. The extrapolation of the linear portion of the sigmoid curve to the abscissa (*f* = 0), and to the highest ordinate value of the fitted plot, affords two values of time (*t*_0_ and *t*_1_), which correspond to the lag time and the end-time reaction, respectively. The time at which half of the protein is aggregated (i.e., when *f* = 0.5) is defined as the time of half-aggregation (*t*_1/2_) [33].

### 2.5. AChE Inhibition Assay

A solution containing 25 µL of acetylthiocholine iodide (ATC) (1.2 × 10^−2^ mol·L^−1^), 250 µL of 5,5′-dithiobis(2-nitrobenzoic acid (DTNB) (2.5 × 10^−2^ mol·L^−1^) and 825 µL of native buffer (NB) (pH 8.0) is prepared. The study sample is obtained by adding 3 µL of flavonoid solution (1 mM) to 57 µL of the previous solution. The reaction starts when 40 µL of AChE (0.08 U mL^−1^) are added to the resulting mixture; the reaction is followed at 412 nm (absorbance) for 2 min. The inhibition (%) of AChE activity is obtained using Equation (2):(2)$$Inh\left(\%\right)=\left\{\left(A_{control}-A_{sample}\right)/A_{control}\right\}\cdot 100\%\;$$
where *Inh*(%) is the inhibitory percentage; *A_sample_* is the absorbance of the sample containing the flavonoid; and *A_control_* is the absorbance of the control, namely the sample without flavonoid.

### 2.6. Fluorescence Quenching Study

The fluorescence emission of AChE was recorded from 300 to 400 nm using an excitation wavelength of 276 nm. The fluorescence intensity was measured at 335 nm. A solution consisting of 40 µL of AChE (0.2 µM) in 100 µL of native buffer (pH 8.0) was used in which 0.5–10 µL of a 10 µM solution of the tested flavonoids was added sequentially. The fluorescence emission was recorded after each addition.

The fluorescence quenching was obtained by applying the Stern–Volmer equation (Equation (3)):(3)$$F_{0}/F=1+K_{q}\tau_{0}\left[Q\right]=1+K_{sv}\left[Q\right]$$
where *F*_0_ and *F* represent the fluorescence intensities in the absence and presence of flavonoid, respectively; [*Q*] is the concentration of the flavonoid tested; *K_q_* is the quenching rate constant; *τ*_0_ is the average lifetime; and *K_sv_* is the Stern–Volmer quenching constant.

The binding constant was determined using Equation (4), which illustrates the relationship between the fluorescence quenching intensity and the concentration of the tested flavonoid.
(4)$$log\{(F_{0}/F)/F\}=logK_{a}+nlog\left[Q\right]$$
where *K_a_* is the binding constant, and *n* is the number of binding sites per AChE molecule. The assays were performed in triplicates, and SD values <5% were obtained in all cases.

### 2.7. Specific Antioxidant Activity Assay

The assay based on 2,2-diphenyl-1-picrylhydrazyl (DPPH) was carried out as described in the literature [34]. Increasing amounts of the polyphenol to be tested were added to a freshly prepared methanolic solution of DPPH (final DPPH concentration of 25 μM). Flavonoid stock solutions were prepared in DMSO. The sample solutions were incubated for 5 min, and the absorbance was recorded from 475 to 525 nm; DPPH indeed exhibits strong absorbance at 517 nm. The percentage of free radical inhibition was calculated using Equation (2) (see above). The assays were performed in triplicates, and SD values <5% were obtained in all cases.

### 2.8. Inhibition Kinetics

The mechanism of AChE enzymatic inhibition was determined using Lineweaver–Burk reciprocal plots. Briefly, a solution (sol A) containing 200 μL of DTNB (1.5 × 10^−3^ mol·L^−1^), 16 µL of AChE (0.5 U·mL^−1^), 9 μL of flavonoid (at the required concentration) and 665 μL of NB were prepared. Then, 200 μL of sol A was mixed with 1, 2, 3 or 4 μL of ATC (15 × 10^−3^ mol·L^−1^) to start the reaction that was followed at 412 nm (absorbance) for 2 min, allowing to obtain the enzymatic ratio.

The results were plotted on a graph of 1/v in μM^−1^·s vs. 1/[ATC] in mM^−1^, and the lines, obtained from the linear regression using Prism software, were extrapolated to determine the intersection points on the abscissa and ordinate axes. The points of intersection allow identifying the type of inhibitor. From these plots, the *K_M_* and *v_MAX_* for each inhibitor were obtained.

### 2.9. Dynamic Light Scattering (DLS) Assays

A solution containing Aβ40 (20 µM), 50 µL of DMSO and 950 µL of native buffer was used. This solution was split; 10 µL of AChE was added to one of the solutions. The two solutions (of 500 µL) were again split, giving four solutions of 250 µL, two of them containing AChE. A 0.5 µL volume of inhibitor (10 mM) was subsequently added to all samples. The samples containing AChE were incubated at 37 °C for 24 h and the controls at 4 °C for 24 h. The particles suspended in liquids (PSD) were measured with DLS.

### 2.10. Fourier-Transform Infrared Spectroscopy (ATR-FTIR)

Attenuated total reflectance–FTIR spectra of lyophilized samples were registered with an FTIR Thermo Scientific Nicolet iS5 spectrometer equipped with a diamond ATR iD7 device. Each spectrum accumulated 64 independent scans, measured at a spectra resolution of 2 cm^−1^ within the range 4000–550 cm^−1^. All spectral data were acquired and baseline-corrected using the Omnic v9.2 software.

### 2.11. Transmission Electron Microscopy (TEM)

Carbon-coated copper grids of 200 mesh were activated through glow discharge for 30 s. Immediately after, the samples were deposited onto the grids, which were thoroughly washed with Milli-Q water. Then, the grids were treated with UranyLess and left to dry in a desiccator for at least 24 h prior to visualization by TEM. The samples were visualized using a Tecnai Spirit TWIN (FEI) 120 kV TEM microscope equipped with a LaB6 emitter and a Megaview 1 k × 1 k CCD.

### 2.12. AChE Preferential Interaction by SDS-PAGE

Aβ40, purchased from Bachem, was dissolved in 50 μL of DMSO at a concentration of 400 µM, and the monomers were solubilized through sonication for 15 min. Native buffer (750 µL) was added, and the sample was divided into four aliquots (190 µL). Then, 0.8 µL of each compound (apigenin or quercetin) at 6 mM in DMSO was added (final concentration of 20 µM) or 0.8 µL of DMSO without compound. Finally, 47.5 µL of AChE (20 µM) was added to obtain a final concentration of 4 µM. The samples were incubated in a thermomixer (Eppendorf, Germany) at 37 °C and stirred at 1400 rpm for 24 h. The negative control is left at 4 °C. The next day, 100 µL of each sample was mixed with 100 µL of Congo Red (CR) at 100 µM and incubated at room temperature for 2 h. After incubation, the samples were precipitated by centrifugation at 16,000× *g* for 60 min, and the soluble fractions were analyzed by Mini-PROTEAN^®^ Tris-Tricine Precast Gels (16.5% bis-acrylamide) (Bio-Rad, Hercules, CA, USA). The gels were stained with Coomassie brilliant blue, scanned at high resolution, and the bands were quantified with GelEval (FrogDance Software v1.37).

### 2.13. Computational Details

The binding mode of quercetin to human AChE (hAChE) was investigated by combining molecular docking, molecular dynamics (MD) simulations, which were used to refine the binding mode, and free-energy calculations, which were carried out to estimate the binding affinity.

Docking was performed using three X-ray structures of hAChE (PDB ID 4M0F, 6CQU and 6O4X; resolution of 2.3 Å) chosen as protein templates in order to consider the distinct arrangement of residues Tyr337 and Trp286 (numbering in the X-ray structures, corresponding to Tyr368 and Trp317 in the UniProt sequence P22303), which are located in the catalytic and peripheral pockets, respectively (Appendix A). Docking of quercetin was performed using the XP score function of Glide [35]. To this end, we defined an inner/outer box of 40/60 Å centered in the gorge at the midpoint between Tyr317 and Trp286, encompassing the catalytic and peripheral sites. The best poses obtained for each template were visually checked, and the best ligand–protein complexes were refined with molecular simulations.

Amber20 was used to run molecular dynamics (MD) simulations [36]. The protein was described using the ff99SB-ILDN force field [37] and was immersed in a TIP3P water box [38] with a layer of 12 Å. The parameterization of quercetin was made using the gaff2 force field [39], and the RESP atomic point charges [40] were determined by fitting the HF/6-31G(d) electrostatic potential. Counterions were adjusted to keep the neutrality of the system and maintain a physiological ionic atmosphere. Then, a three-step minimization protocol was used to gradually minimize hydrogen atoms, water molecules and finally the whole system. The protein–ligand systems were then heated to 300 K in six steps imposing a Cartesian restraint of 5 kcal mol^−1^ Å^−2^ to preserve the binding mode of quercetin, avoiding the occurrence of artefactual changes during equilibration, at constant pressure (1 bar) conditions. Starting from the equilibrated structures, three replicas (R1-3) were run for each system. Production calculations were performed at constant volume and temperature using SHAKE [41] for bonds involving hydrogen atoms. Electrostatic interactions were treated with particle mesh Ewald (PME) [42], and a cut-off of 10 Å was established for the nonbonded interactions. The restraints were gradually removed in the first 25 ns. Finally, all replicas were run for 300 ns.

The binding mode was assessed by examining the structural stability of the ligand, and the binding affinity was estimated using the Amber package MMPBSA.py. MM-GBSA [43] was used to estimate the different contributions to the free energy (Equation (5)). Since the major aim is to compare distinct binding modes of quercetin, the loss of entropic contribution due to ligand binding was assumed to cancel for the three binding modes and was not explicitly considered.
(5)$$G=E_{int}+E_{el}+E_{vw}+G_{sol,el}+G_{sol,n-el}-T\Delta S$$

In Equation (5), $E_{int}$, $E_{el}$ and $E_{vw}$ stand for the internal, electrostatic and van der Waals energy terms, respectively; $G_{sol,el}$ is the electrostatic contribution to the solvation free energy, which was determined using the generalized Born solvation method; and $G_{sol,n-el}$ is the nonpolar contribution to the solvation free energy, which was computed from the solvent-accessible surface area (SASA).

The binding affinity was determined using Equation (6), and the free energy of each species (complex, receptor and ligand) was determined for an ensemble of 50 snapshots taken along the last 50 ns (chosen due to the stability of the RMSD profiles; see below) of the trajectories run for the protein–ligand complex.
(6)$$\Delta G_{bind}=G_{complex}-G_{receptor}-G_{ligand}$$

## 3. Results and Discussion

### 3.1. Inhibition of Aβ40 Aggregation

Aβ40 fibrillation was tracked by fluorescence using the specific amyloid dye ThT, which binds to β-sheet-rich structures giving rise to an increase in the fluorescence. As illustrated in Figure 2, ThT fluorescence is strong in the presence of amyloid-like aggregates (green spectrum), whereas it is low in the presence of nonfibrillated Aβ (black dotted spectrum). Remarkably, the ThT signal is drastically reduced in the presence of quercetin (red spectrum) and apigenin (blue spectrum), suggesting that the presence of these compounds prevents the formation of fibrils (Figure 2). Quercetin is the most effective compound with 85.1% inhibition when the Aβ/quercetin ratio is 1:1, whereas apigenin leads to 74.8% inhibition at an equimolar concentration. This finding may be explained by the ability of flavonoids to intercalate between Aβ peptides and form hydrogen bonds with the peptide backbone, hence acting as β-sheet disruptors [30]. The slight difference in inhibitory activity exhibited by the two flavonoids might reflect the formation of additional interactions arising from the presence of more hydroxyl groups in quercetin (Figure 1).

To further study the interaction of these flavonoids with Aβ40, time-course kinetics experiments were carried out in the presence and absence of quercetin and apigenin. The ThT fluorescence data confirm the inhibitory activity of both compounds, the larger inhibition being triggered by quercetin (Figure 3 and Table 1).

Compared to free Aβ, the nucleation rate constant (*k*_n_) is increased by a factor of 3.2 and 4.5 upon incubation with apigenin and quercetin, respectively (Table 1). This suggests the occurrence of favorable interactions between the compounds and Aβ40 in the first steps of the nucleation, altering the normal fibrillation pathway, as noticed for other systems (i.e., with nanoparticles), and may lead to the generation of aggregates in nonamyloid conformations [44]. In agreement with these kinetic observations, a very recent study has also concluded that the early formation of Aβ40/quercetin complexes entails an increase in the concentration of nonproductive oligomers (off-pathway Aβ-quercetin oligomers) as a potential mechanism for quercetin action [45]. Interestingly, the elongation rate (*k*_e_) is not affected in the case of apigenin, or reduced by a factor of two in the case of quercetin. The presence of these flavonoids also reduces the lag time *t*_0_, this effect being more pronounced for quercetin. Conversely, *t*_1/2_ and *t*_1_ are reduced in the presence of apigenin, whereas they are increased with quercetin.

### 3.2. Inhibition of Aβ40 Aggregation Induced by AChE

AChE contains an α/β hydrolase fold with the catalytic active site (CAS) located at the end of a deep (ca. 20 Å long) gorge. A peripheral anionic site (PAS) is found at the entrance of the gorge and modulates the entry of small molecules, viz. substrates or inhibitors [46,47]. Since AChE accelerates Aβ40 aggregation and facilitates the formation of Aβ plaques [15,16,17,18,19,48], research efforts have been devoted to designing small compounds, including flavonoids, that target the catalytic and peripheral sites of AChE [25,49,50,51,52].

As shown in Figure 4, the AChE-induced fibrillation of Aβ40 is reduced by 30.3 and 55.4% in the presence of apigenin and quercetin, respectively. The inhibition triggered by the two flavonoids on the AChE-induced aggregation of Aβ is lower compared with the inhibitory activity determined in the absence of the enzyme (see Figure 1 and Figure 3). This difference can be attributed to the binding of the flavonoids to AChE, thus reducing the effective amount available to interact with Aβ40. Since the AChE-induced Aβ40 aggregation reflects the formation of Aβ40/AChE complexes, we performed a centrifugation assay of the aggregated Aβ40/AChE samples in the presence and absence of apigenin and quercetin to further check the effect of these flavonoids on the formation of Aβ40 fibrils. The supernatant (containing soluble AChE) was separated from the pellet (containing AChE coprecipitated with Aβ40 fibrils) and was analyzed by SDS-PAGE. Under reducing conditions, *Electric eel* AChE (EeAChE) appears as two main bands comprising between 50 and 75 kDa (Appendix A), which agrees with the results reported in previous studies [53,54,55]. The concentration of soluble AChE is drastically reduced when Aβ40 is aggregated in the absence of inhibitors due to the interaction between Aβ40 and AChE. In the presence of apigenin, a large reduction of the bands is also observed. In contrast, in the presence of quercetin, the two main bands typical of soluble AChE can be noticed, albeit showing a slight reduction of 15%, suggesting that binding of quercetin to AChE seems to be more effective in preventing the AChE-induced Aβ40 aggregation.

Time-course kinetics experiments of the AChE-induced Aβ40 aggregation were subsequently carried out. Compared to the data obtained without AChE (Table 1), the rate of the nucleation process is reduced by a factor of ~2 in the presence of AChE, but this is counterbalanced by a similar increase in the elongation rate constant, which is accompanied by a decrease in the half-aggregation time (*t*_1/2_) (Table 2). Incubation with quercetin and apigenin increases the value of *k*_n_ by 2.4- and 3.3-fold, respectively (Table 2 and Figure 5), thus showing an attenuation of the effect relative to the absence of AChE (it can be noted that *k*_n_ increased by 3.2-and 4.5-fold in the presence of apigenin and quercetin, respectively; Table 1). This feature can be attributed to the binding of apigenin and quercetin to AChE (see below). However, the elongation constant *k*_e_ remains mostly unaltered, suggesting that apigenin and quercetin interact mainly with species formed at the early stages of the aggregation. Inspection of Table 2 shows that quercetin gives rise to a larger change in the kinetic parameters obtained for the AChE-induced aggregation of Aβ40 compared to apigenin, which may reflect differences in the interaction of the two flavonoids with AChE.

### 3.3. Structure of Aβ40 and AChE-Induced Aβ40 Aggregates

Transmission electron microscopy (TEM), dynamic light scattering (DLS) and Fourier-transform infrared spectroscopy (FTIR) were used to gain insight into the inhibitory effect of quercetin and apigenin on Aβ aggregation. The TEM images show clear macroscopic differences between the different samples (Figure 6). Comparison between the flavonoid-free aggregation of Aβ in the absence or presence of AChE reveals that the fibers are structurally analogous. However, they tend to assemble in the presence of AChE (Figure 6, top). Significant alterations of the quaternary fibril structure occur upon incubation with the two flavonoids (with and without AChE present; Figure 6, middle and bottom). Thus, besides the presence of amorphous species, the fibrillar structures seem to be shorter when the flavonoids are present. It can be pointed out that the presence of amorphous aggregates agrees with the reduction of the ThT signal observed in Figure 1 and Figure 3, confirming the inhibitory effect exerted by flavonoids on Aβ aggregation.

DLS measurements were performed to determine the final size of aggregates. Although this technique only allows the analysis of structures of discrete size, it can perceive structural changes at the macroscopic level [56,57]. Interestingly, the fibrils formed in the presence of AChE appear to be half the size of those generated without enzymes (Table 3). The flavonoids affect the Aβ aggregation both in the presence and absence of AChE. In this latter case, there is a size reduction of 25 and 50% for apigenin and quercetin, respectively (Table 3). In contrast, with AChE, the size of aggregates is practically not altered (only a small reduction is observed in the presence of quercetin), indicating that the macromolecular structure of the fibrils is not noticeably changed by the flavonoids.

Finally, the secondary structure of the aggregates obtained under the different experimental conditions was investigated by ATR-FTIR. After deconvolution of the amide I region of the spectra, two important peaks are observed at 1611 and 1631 cm^−1^ (Table 4), which are assigned to inter- and intramolecular β-sheet structures, respectively [58]. All the FTIR patterns (shown in Figure 7) are highly similar, supporting the presence of similar secondary structure content in all conditions.

### 3.4. Inhibition of AChE Activity by Apigenin and Quercetin

The AChE activity was determined using the well-known spectrophotometric Ellman’s assay using several enzymatic concentrations [59]. As illustrated in Figure 8 and Appendix A, the enzymatic activity decreases when the concentration of flavonoid is increased. Half-maximal inhibitory concentrations (IC_50_) were found to be 52.9 and 40.7 µM for apigenin and quercetin, respectively, which suggests a slightly stronger interaction of quercetin with the enzyme, most likely due to the additional hydroxyl groups (Figure 1). Lineweaver–Burk plots (Figure 9) point out that the flavonoids are noncompetitive inhibitors, as they trigger a reduction in V_max_ (free enzyme: 3.94 μM·s^−1^; with apigenin: 1.28 μM·s^−1^; with quercetin: 1.74 μM·s^−1^), but have little impact on the *K*_M_ (free enzyme: 0.15 μM; with apigenin: 0.15 μM·s^−1^; with quercetin: 0.20 μM·s^−1^).

### 3.5. Fluorescence Quenching of AChE

The interaction of the flavonoids with AChE was investigated through the quenching of the fluorescence of AChE. The binding of both flavonoids to AChE substantially quenches its fluorescence, as noted in the change observed in the fluorescence spectra of AChE in the presence of increasing amounts of apigenin and quercetin (Appendix A, respectively). These findings suggest that apigenin and quercetin may bind to pockets containing Trp residues, such as Trp86 in the catalytic site and Trp286 in the peripheral site (Appendix A). However, the noncompetitive mechanism observed for the AChE inhibition exerted by the two compounds supports the binding to the peripheral site (see above).

As illustrated in Figure 10a, the intensity of AChE fluorescence decreases rapidly when apigenin or quercetin are added, this effect being slightly larger for apigenin. The quenching effect of the two flavonoids was estimated using the Stern–Volmer equation (Figure 10b), and the binding affinities of quercetin and apigenin for AChE were then estimated from the plots log{(F_0_/F)/F} vs. log[flavonoid] (Figure 10c). Both flavonoids bind to AChE in a 1:1 stoichiometry, the affinity constant for apigenin being 4-fold higher than that of quercetin (Table 5). The data suggest that apigenin would be more effective than quercetin in blocking the effect of AChE on amyloid formation, in agreement with the findings reported by Inestrosa and coworkers [17,18]. Furthermore, this would also justify the larger resemblance observed for the kinetic parameters determined for the AChE-induced aggregation of Aβ40 in the absence and presence of apigenin (see Table 2).

### 3.6. Computational Study of the Binding Mode of Quercetin to AChE

To explore the molecular basis of the AChE inhibition, the binding mode of quercetin to AChE was investigated by combining molecular docking and MD simulations. Although the AChE inhibitory activity was experimentally determined using the *Electrophorus electricus* enzyme (eeAChE), computational studies were performed considering the X-ray crystallographic structures of hAChE deposited in the Protein Data Bank with codes 4M0F, 6CQU and 6O4X. This choice was motivated by several reasons. First, the lack of crystallographic structures for the eeAChE enzyme, since the low resolution achieved in previous studies impeded the modeling of the details of the backbone and side chains [53]. Second, the large preservation of the residues that shape the catalytic and peripheral binding sites suggests that the ligand will adopt a similar binding mode in both eeAChE and hAChE. For instance, only three differences can be noticed for the residues in the peripheral site: two correspond to two conserved replacements (Ile267 and Leu289 in eeAChE by V282 and Ile294 in hAChE, respectively), and the third involves the change of Gly288 in eeAChE by Ser293 in hAChE, but this substitution is located on the edge of the binding pocket. Third, the large number of ligand complexes available with hAChE permits the identification of ligand-adapted conformational changes in certain residues, such as Tyr337 in the catalytic pocket and Trp286 in the peripheral site, which can be found in distinct arrangements (Appendix A), as was noticed in earlier studies [60,61]. Keeping in mind the sensitivity of docking to the positional details of residues, it is then convenient to perform these calculations considering the X-ray structures 4M0F, 6CQU and 6O4X as templates in order to explore the effect of the conformational rearrangements of Tyr337 and Trp286 on the binding pose of quercetin.

The docking results pointed out a preferential binding of quercetin at the peripheral site in all cases, excluding the feasibility of binding at the catalytic site. Remarkably, this agrees with the noncompetitive inhibition mechanism found for this compound (see above) and previous studies that reported that only PAS inhibitors can block the AChE-induced aggregation [17,18,19].

The score of the top-ranked poses ranged from −11.9 to −6.1 kcal∙mol^–1^. The best pose was obtained in the docking to 4M0F and 6O4X (−11.9 and −10.6 kcal∙mol^–1^, respectively), likely reflecting the formation of *π*–*π* stacking interactions between quercetin and both Trp286 and Tyr341. In contrast, an outer pose was found in the docking to 6CQU (−6.1 kcal∙mol^–1^), though still retaining contact with Trp286. In line with these results, the top-ranked binding modes obtained for 4M0F, 6O4X and 6CQU were refined by means of MD simulations. The RMSD profile obtained for the protein backbone was stable along the trajectories, and only minor adjustments were observed for the residues that shape the peripheral site, as expected from the solvent exposure of this binding pocket (Appendix A). However, the RMSD profile of the ligand often reflected structural rearrangements in the first 50 ns, and then remained stable till the end of the trajectory. The final arrangement adopted by the ligand at the peripheral site is shown in Figure 11.

For 6CQU, two replicas (Figure 11a,b) exhibited a similar binding mode characterized by *π*–*π* stacking interactions with Trp286 and Tyr124, which was transiently assisted by the formation of a hydrogen bond between the hydroxyl group in Position 7 of the 4H-chromen-4-one moiety and Glu285, and between the hydroxyl group at Position 4 and Asp74. In the simulations run for 4M0F complexes, two putative binding modes were found for quercetin (Figure 11c,d). In both configurations, the 4H-chromen-4-one ring is exposed to the solvent, mainly due to the different arrangement of Trp286, which forms a *π*–*π* interaction. Moreover, quercetin is hydrogen-bonded to the backbone units of Phe338 and Pro290. Finally, for 6O4X runs, only one binding mode was retained (Figure 11e). Again, Trp286, with a different orientation with respect to the other cases, forms a stacking interaction with quercetin, which is further assisted by the stacking with Tyr341 and hydrogen bonds with Tyr77 and Thr83.

The contributions to the binding affinity determined for these poses are shown in Table 6. Broadly speaking, the estimated ∆*G_bind_* values indicate that 6CQU—Replica 1 (−21.0 kcal∙mol^–1^) and 4M0F—Replica 3 (−21.3 kcal∙mol^–1^) are the most stable complexes, followed by 6CQU—Replica 2 (−19.6 kcal∙mol^–1^), 4M0F—Replica 2 (−17.2 kcal∙mol^–1^) and 6O4X—Replica 3 (−16.6 kcal∙mol^–1^). This reflects the potential capabilities conferred by the hydroxyl groups attached to the core of this flavonoid, suggesting that the interaction of quercetin at the peripheral site may involve more than a single binding mode.

### 3.7. Radical Scavenging Properties of the Flavonoids

The antioxidant properties of some antiamyloid agents may represent a beneficial factor, contributing to their global antiaggregation activities. The radical scavenging abilities of apigenin and quercetin were thus studied using DPPH. As illustrated in Figure 12b, the absorbance of DPPH at 517 nm decreases with the addition of increasing amounts of quercetin. In contrast, no effect was observed with apigenin (Figure 12a). Quercetin concentrations in the low micromolar range (≤3 μM; Table 7) were able to totally quench the absorbance of the DPPH radical (Figure 12b), suggesting that quercetin may be capable of reducing the oxidative stress, namely by eliminating ROS.

## 4. Conclusions

The results obtained in the present study reveal the effects exerted by flavonoids apigenin and quercetin on the interplay between (i) the inhibition of the enzymatic activity of AChE, (ii) the formation of Aβ40 fibrils and (iii) the AChE-promoted Aβ aggregation.

The reduction in the formation of Aβ fibrils observed in the presence of quercetin and apigenin is reflected by the increase in the nucleation rate constant and the decrease in the lag time t_0_ relative to free Aβ, suggesting the occurrence of favorable interactions between these compounds and Aβ40 in the first steps of the aggregation, altering the normal fibrillation pathway. The presence of AChE leads to a notable reduction of the time required to complete the Aβ fibrillation; the analysis of the kinetics parameters reflects that the reduction of the nucleation process is counterbalanced by an increase in the elongation rate constant, leading to a decrease in the half-aggregation time. Incubation with apigenin and quercetin alters the influence exerted by AChE on Aβ40 aggregation, leading to an inhibition of 25.5 and 57.9%, respectively. Nevertheless, the extent of this effect is clearly mitigated compared with the antiaggregating activity of these compounds in the absence of AChE, as the inhibition was 66.5 and 74.9% for apigenin and quercetin, respectively.

The reduction of the antiamyloid activity of the flavonoids in the presence of AChE is indicative of interactions between the flavonoids and AChE. The noncompetitive mechanism observed in the enzymatic inhibition assays, together with the decrease in the fluorescence of the enzyme (presumably due to the interaction with Trp286), agrees with an interaction of apigenin and quercetin at the peripheral site of the enzyme, as also shown by molecular modeling studies, which reveal the interaction with Trp286. Noteworthy, the lower association constant determined for the binding of quercetin to AChE provides a basis to justify the larger effect observed on the kinetics of the AChE-induced aggregation.

Overall, these findings suggest that a proper choice of the chemical groups that decorate the flavonoid scaffold can be exploited to finely tune the balance between the AChE inhibition, Aβ40 fibrillation, and AChE-Aβ interaction. In conjunction with the antioxidant properties (relief of oxidative stress) and the ability to reach the central nervous system [62], flavonoids with a suitably decorated scaffold might be used as coadjutant agents or be incorporated in multitarget compounds for the treatment of AD.

## Figures and Tables

**Figure 1 pharmaceutics-14-02342-f001:**
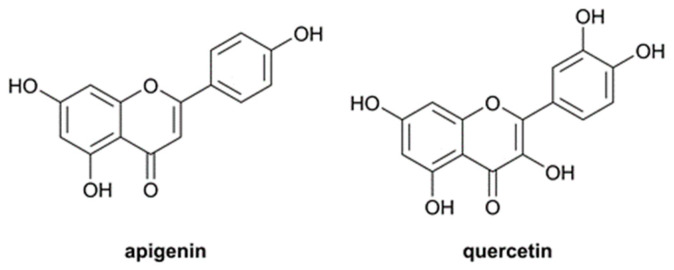
Representations of the chemical structures of apigenin and quercetin.

**Figure 2 pharmaceutics-14-02342-f002:**
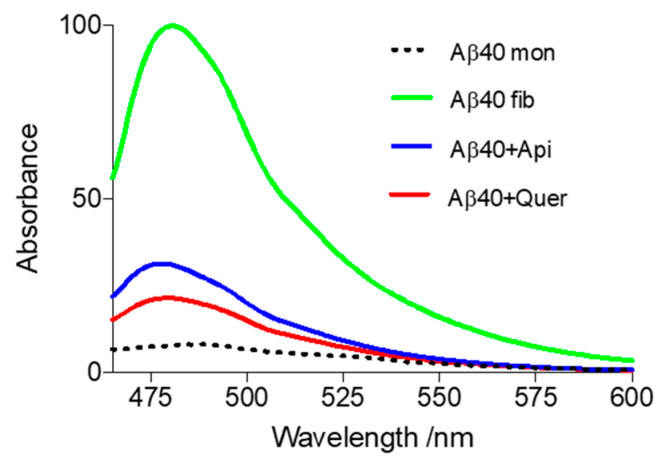
ThT fluorescence in the presence of Aβ40: Effect of aggregation inhibitors. The samples were incubated at 37 °C for 24 h and fluorescence emission spectra were recorded in the range 465—600 nm. Green spectrum: typical aggregation of Aβ; black dotted spectrum: Aβ monomers incubated at 4 °C (negative control); red spectrum: aggregation in the presence of quercetin; blue spectrum: aggregation in the presence of apigenin. Conditions: [Quercetin/Apigenin] = 20 µM; [Aβ40] = 20 µM. The assays were performed in triplicates, and SD values <5% were obtained in all cases (not shown for the sake of clarity).

**Figure 3 pharmaceutics-14-02342-f003:**
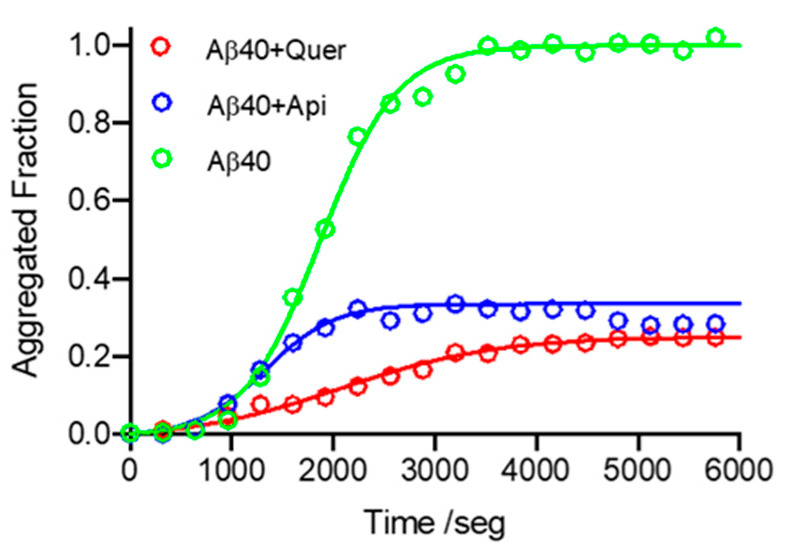
Time-course kinetics of Aβ40 aggregation in the absence and presence of quercetin and apigenin, followed by ThT fluorescence. Green: without flavonoid; blue: with apigenin; red: with quercetin. Conditions: [Quercetin/Apigenin] = 20 µM; [Aβ40] = 20 µM.

**Figure 4 pharmaceutics-14-02342-f004:**
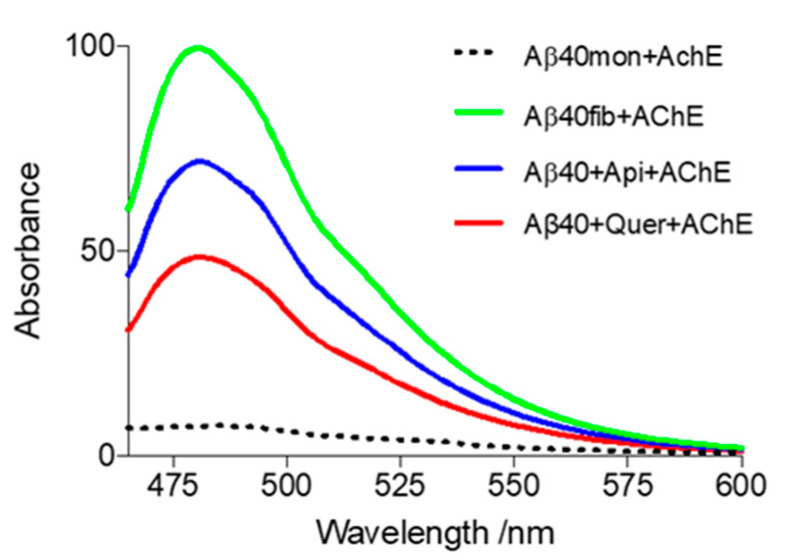
Effect of quercetin and apigenin on the AChE-induced fibrillation of Aβ40. The samples were incubated at 37 °C for 24 h and ThT fluorescence emission spectra were recorded in the range 465–600 nm. Green spectrum: typical aggregation of Aβ; black dotted spectrum: Aβ monomers incubated at 4 °C (negative control); red spectrum: aggregation in the presence of quercetin; blue spectrum: aggregation in the presence of apigenin. Conditions: [Quercetin/Apigenin] = 20 µM; [Aβ40] = 20 µM; [AChE] = 4 μM. The assays were performed in triplicates, and SD values <5% were obtained in all cases (not shown for the sake of clarity).

**Figure 5 pharmaceutics-14-02342-f005:**
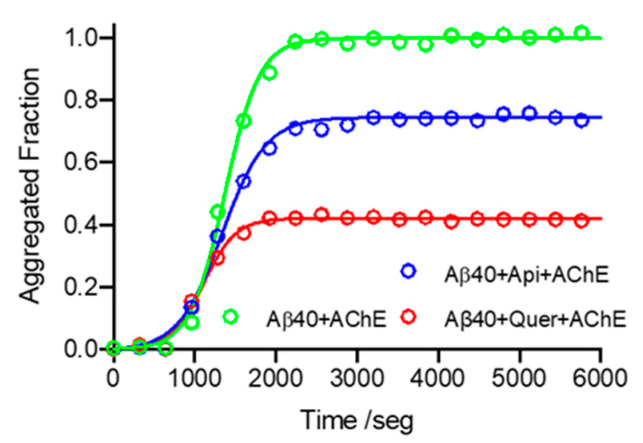
Time-course kinetics of AChE-induced Aβ40 aggregation in the absence and presence of the two flavonoids quercetin and apigenin, followed by ThT fluorescence. Green: without flavonoid; blue: with apigenin; red: with quercetin. Conditions: [Quercetin/Apigenin] = 20 µM; [Aβ40] = 20 µM.

**Figure 6 pharmaceutics-14-02342-f006:**
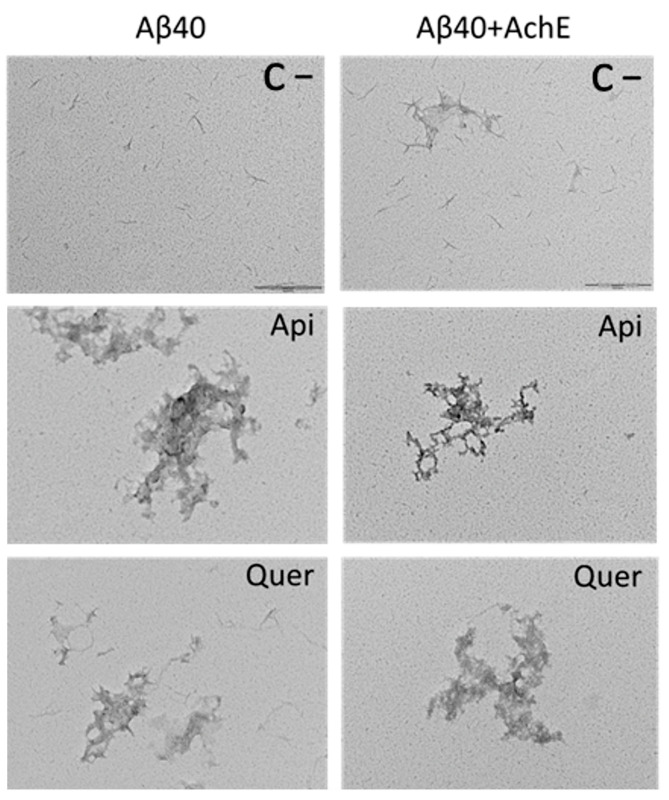
TEM images of Aβ40 fibrils generated without (**left**) and with (**right**) AChE, and in the absence (**top**) or presence of the flavonoids apigenin (Api; **middle**) and quercetin (Quer; **bottom**). Scale bars: 200 nm.

**Figure 7 pharmaceutics-14-02342-f007:**
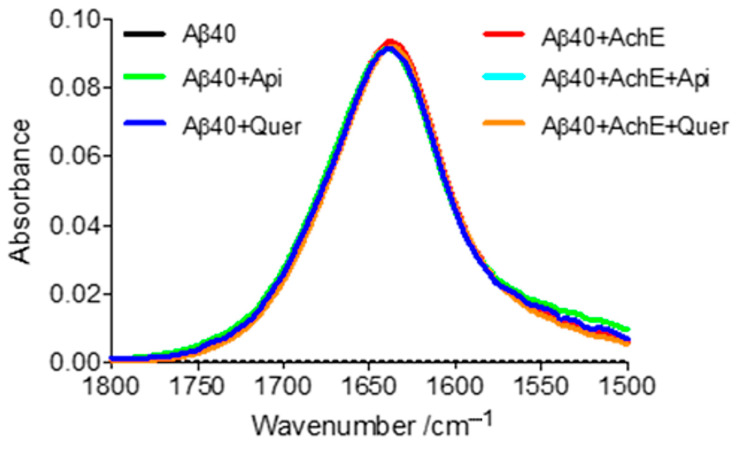
FTIR patterns centered in the amide I region of Aβ40 for different samples after 24 h of incubation. Black spectrum: free Aβ; green spectrum: Aβ + apigenin; dark-blue spectrum: Aβ + quercetin; red spectrum: Aβ + AChE; light-blue spectrum: Aβ + AChE + apigenin; orange spectrum: Aβ + AChE + quercetin.

**Figure 8 pharmaceutics-14-02342-f008:**
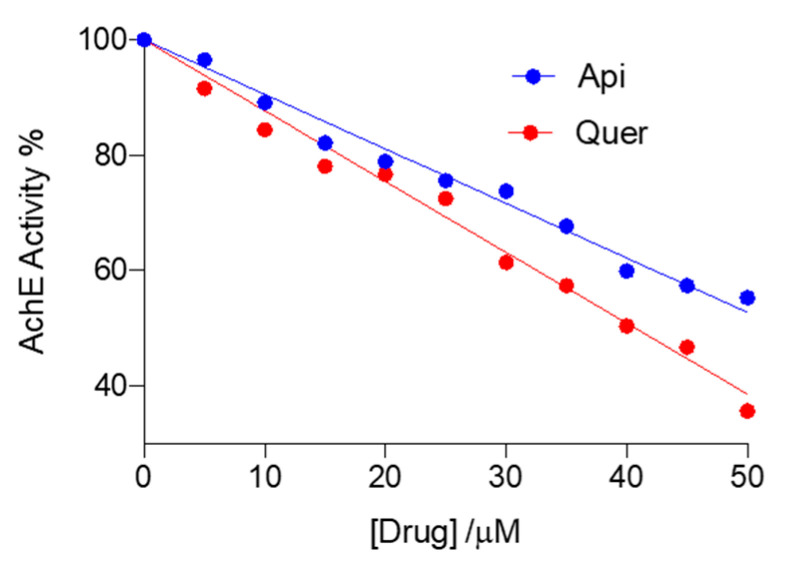
AChE activities as a function of the concentration of flavonoid used, measured spectrophotometrically at 412 nm. The concentrations of apigenin (Api) and quercetin (Quer) flavonoids, i.e., [Drug], range from 0 to 50 μM; [AChE] = 0.08 U mL^−1^.

**Figure 9 pharmaceutics-14-02342-f009:**
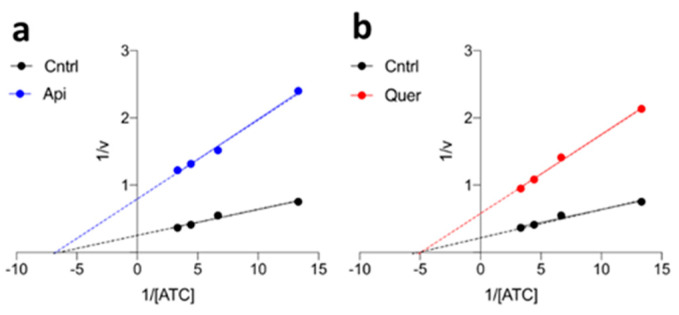
Lineweaver–Buck plots. (**a**) Apigenin. (**b**) Quercetin.

**Figure 10 pharmaceutics-14-02342-f010:**
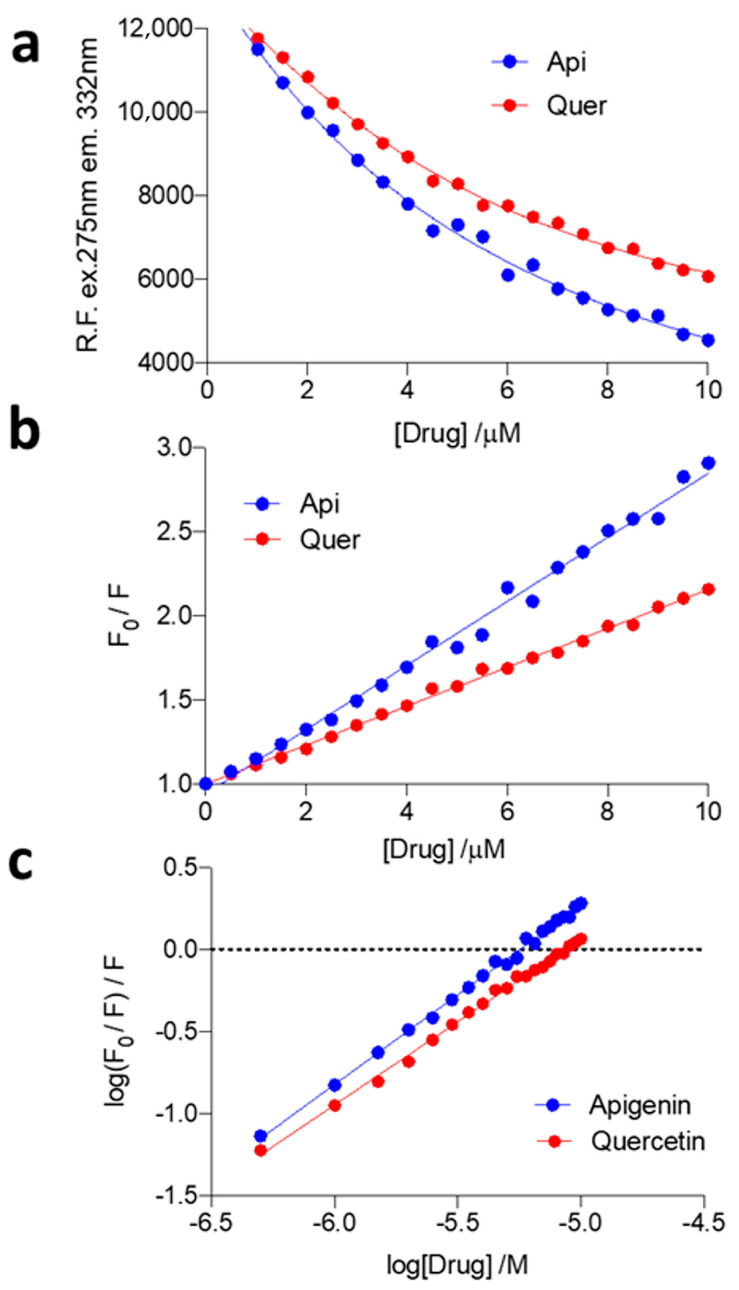
Fluorescence quenching of AChE by quercetin and apigenin (Drug in the plots). (**a**) Intrinsic fluorescence emission of AChE upon addition of increasing amounts of quercetin (red) and apigenin (blue). (**b**) Stern–Volmer plots, viz. F_0_/F vs. [flavonoid] showing the quenching of AChE fluorescence in the presence of increasing amounts of quercetin (red) and apigenin (blue). (**c**) Relationship between the binding affinities and the numbers of binding sites for the interaction between AChE and apigenin (blue) and between AChE and quercetin (red). λ_exc_ = 275 nm, λ_em_ = 332 nm.

**Figure 11 pharmaceutics-14-02342-f011:**
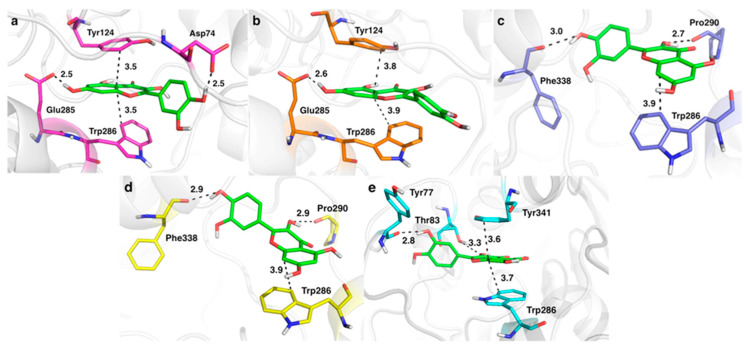
Five selected binding modes for quercetin (carbon atoms in green) with different human AChE systems. (**a**) 6CQU—Replica 1 (carbon atoms in magenta), (**b**) 6CQU—Replica 2 (carbon atoms in orange), (**c**) 4M0F—Replica 2 (carbon atoms in light blue), (**d**) 4M0F—Replica 3 (carbon atoms in yellow) and (**e**) 6O4X—Replica 3 (carbon atoms in cyan). Selected residues in the binding pocket are indicated as bold text. Dashed lines represent interatomic distance (values in Å).

**Figure 12 pharmaceutics-14-02342-f012:**
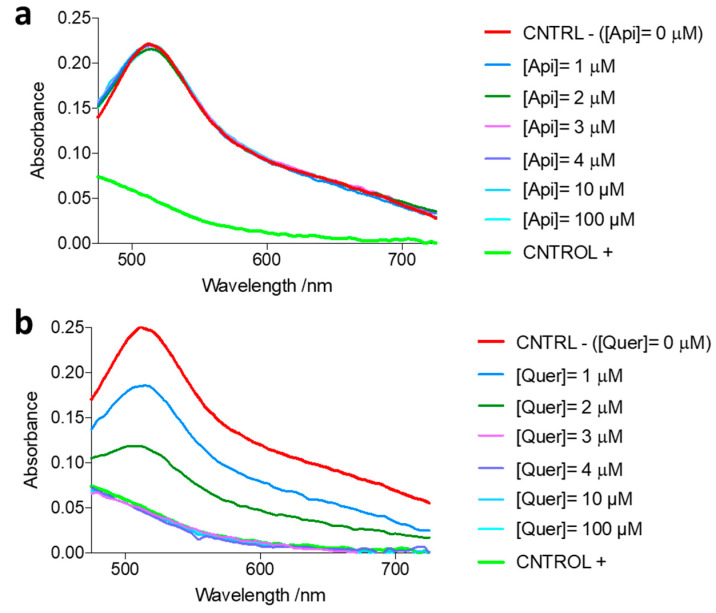
Radical scavenging abilities of (**a**) apigenin and (**b**) quercetin, determined through the absorbance of DPPH at 517 nm, using increasing concentrations of the flavonoids.

**Table 1 pharmaceutics-14-02342-t001:** Kinetic parameters obtained for the aggregation of free Aβ40, and upon incubation with apigenin and quercetin.

	Aβ40	Aβ40 + Api	Aβ40 + Quer
*k*_n_ (10^−5^ s^−1^)	1.83	5.88	8.27
*k*_e_ (M^−1^·s^−1^)	175.73	191.27	80.93
*t*_0_ (s)	1060.4	641.0	321.2
*t*_1/2_ (s)	1844.1	1315.3	2182.1
*t*_1_ (s)	2627.7	1989.6	4043.0
Inhibition (%)	0.0	66.5	74.9

Note: [Aβ40] = 20 µM; [Quercetin/Apigenin] = 20 μM.

**Table 2 pharmaceutics-14-02342-t002:** Kinetic parameters obtained for the AChE-induced aggregation of free Aβ40, Aβ40 with apigenin and Aβ40 with quercetin.

	Aβ40 + AChE	Aβ40 + AChE + Api	Aβ40 + AChE + Quer
*k*_n_ (10^−5^ s^−1^)	0.80	2.57	1.92
*k*_e_ (M^−1^·s^−1^)	309.0	246.2	333.7
*t*_0_ (s)	868.6	736.8	481.9
*t*_1/2_ (s)	1360.0	1323.2	1096.2
*t*_1_ (s)	1851.3	1909.6	1710.6
Inhibition (%)	0.0	25.5	57.9

Note: [Aβ40] = 20 µM; [Quercetin/Apigenin] = 20 μM.

**Table 3 pharmaceutics-14-02342-t003:** Size distributions and polydispersity (PDI) of Aβ40 fibrils generated without and with AChE, and in the absence or presence of apigenin and quercetin, determined by dynamic light scattering (DLS).

	z-Average dm./nm	PDI
Aβ40	3074	0.59
Aβ40 + Apigenin	2399	0.56
Aβ40 + Quercetin	1402	0.46
Aβ40^AChE^	1547	0.42
Aβ40^AChE^ + Apigenin	1585	0.58
Aβ40^AChE^ + Quercetin	1351	0.60

**Table 4 pharmaceutics-14-02342-t004:** Absorption peaks characterizing the secondary structure of Aβ40, determined by FTIR spectroscopy.

Peak	Peak (cm^−1^)	Area (%)
1	1611.1	31.4
2	1631.0	24.6
3	1650.1	6.1
4	1668.1	37.8

**Table 5 pharmaceutics-14-02342-t005:** Binding affinities (*K*_a_) and numbers of binding sites (*n*) for the interaction of apigenin and quercetin with AChE.

	AChE + Apigenin	AChE + Quercetin
*K*_a_ (10^6^ M^−1^)	0.53	0.13
*n*	1.09	1.01

**Table 6 pharmaceutics-14-02342-t006:** Binding affinities (*K*_a_) and numbers of binding sites (*n*) for the interaction of apigenin and quercetin with AChE.

Models	Δ*E_vw_*	Δ*E_el_*	Δ*G_sol,el_*	Δ*G_sol-n-el_*	Δ*G_bind_*
6CQU—Replica 1	−35.2 ± 2.1	−41.7 ± 4.9	−41.7 ± 4.9	−41.7 ± 4.9	−21.0 ± 2.5
6CQU—Replica 2	−36.0 ± 2.2	−25.8 ± 4.3	−25.8 ± 4.3	−25.8 ± 4.3	−19.6 ± 2.8
4M0F—Replica 2	−34.0 ± 2.2	−24.7 ± 6.8	−24.7 ± 6.8	−24.7 ± 6.8	−17.2 ± 3.0
4M0F—Replica 3	−37.1 ± 1.9	−25.9 ± 4.3	−25.9 ± 4.3	−25.9 ± 4.3	−21.3 ± 2.5
6O4X—Replica 3	−33.8 ± 2.6	−44.6 ± 8.3	−44.6 ± 8.3	−44.6 ± 8.3	−16.6 ± 2.9

**Table 7 pharmaceutics-14-02342-t007:** Radical scavenging capacity (RSC, in %) of apigenin and quercetin, determined by their ability to quench the absorbance of the DPPH radical.

[Quercetin/Apigenin]/μM	Apigenin RSC (%)	Quercetin RSC (%)
1	−0.1	30.2
2	1.8	65.1
3	0.2	99.3
4	2.4	100.9
10	−0.7	100.1
100	2.9	99.7

Note: Standard deviation <5%.

## Data Availability

Not applicable.

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
