# Peer review of "Three to Tango: Inhibitory Effect of Quercetin and Apigenin on Acetylcholinesterase, Amyloid-β Aggregation and Acetylcholinesterase-Amyloid Interaction"

_pharmaceutics, 2022, doi:10.3390/pharmaceutics14112342_

Round 1

Reviewer 1 Report

The manuscript entitled “Three to Tango: Inhibitory Effect of Quercetin and Apigenin on Acetylcholinesterase, Amyloid-β Aggregation and Acetylcholinesterase-Amyloid Interaction” were assessed. In this study, the authors have provided a complete analysis of the potential AChE inhibition and Aβ anti-aggregating activity of quercetin and apigenin. They also investigated the possible effect of AChE in promoting the formation of Amyloid-β Aggregation. Furthermore, the effects of quercetin and apigenin on the interaction between AChE and Aβ are studied. Almost all aspects of the inhibitory activity of quercetin and apigenin against the AChE enzyme are studied well. The work comprehensively includes in-vitro assay, analysis of amyloid aggregation, AChE inhibitory study, fluorescence quenching study, antioxidant activity assay, kinetic studies, DLS assay, FT-IR spectroscopy, TEM analysis, AChE preferential interaction by SDS-PAGE, and computational simulations (docking studies). All sections are well explained. The findings are important to be worthy of attention and the results are significant for new anti-Alzheimer drug designs. So, the work has the qualities that deserve publication.

Author Response

We appreciate the positive remarks made by the reviewer.

Reviewer 2 Report

This work presents in-dept anti-beta amyloid aggregation mechanism of the flavonoids quercetin and apigenin. In vitro and in silico techniques were appropriated. The findings were presented with clarity. This research would have an impact on the use of flavonoids as anti-Alzheimer agents.    

Author Response

The favorable remarks made by reviewer 2 are highly appreciated.

Reviewer 3 Report

Nowadays, there are not effective treatment of Alzheimer’s disease (AD) and this disorder is very complex and many mechanisms are involved in the neurodegeneration processes. AD is the most known form of dementia and a progressive neurodegenerative disease. Therefore, research to discover a new potential anti-AD agent are valuable.

Cholinesterase inhibitors are among the oldest drugs against AD, but they are only symptomatic treatment. Despite this, they are still the subject of research, as the cholinergic system is involved in memory processes. Aggregation of beta-amyloid is also associated with this disease.

The presented manuscript entitled “Three to Tango: Inhibitory Effect of Quercetin and Apigenin on Acetylcholinesterase, Amyloid-β Aggregation and Acetylcholinesterase-Amyloid Interaction” present interesting studies concerning inhibitory properties of acetylcholinesterase enzyme,   fibrillation of beta amyloid and interactions between enzyme-beta-amyloid. Two antioxidants apigenin and quercetin were a subject of investigations.

I propose to correct the manuscript by adding some additional information and explanations.

On the page 2 authors write:

“the proteolytic cleavage of the amyloid precursor protein (APP) generates amyloid-β (Aβ) peptides characterized by variable sequences of 36−43 amino acids. The length of the cleaved fragments affects both cell toxicity and aggregation, the most fibrillogenic 42 amino acid-long peptide (Aβ42) being mainly found in the deposits observed in AD patients.”

In the conducted experiments, beta-amyloid 40 was used. A comment is needed.

For molecular modelling analysis human AChE  was investigated using three X-ray structures of hAChE.

Acetylcholinesterase from Electrophorus electricus was used to evaluate the inhibitory properties of the enzyme in in vitro studies. A comment is needed.

Author Response

We acknowledge the positive comments made by reviewer 3. With regard to the two specific points raised by the reviewer (shown here as text in italics), the manuscript has been modified to include additional comments (see below).

1) On the page 2 authors write:

“the proteolytic cleavage of the amyloid precursor protein (APP) generates amyloid-β (Aβ) peptides characterized by variable sequences of 36−43 amino acids. The length of the cleaved fragments affects both cell toxicity and aggregation, the most fibrillogenic 42 amino acid-long peptide (Aβ42) being mainly found in the deposits observed in AD patients.”

In the conducted experiments, beta-amyloid 40 was used. A comment is needed.

The text has been modified as follows:

'... (or fibrillation). [12] Aβ40 and Aβ42 are the main components of the senile plaques. Since anti-amyloid drugs tested in vitro show very similar effects for both peptides, the physicochemical properties of Aβ40 (viz. higher solubility and lower aggregation propensity relative to Aβ42) makes this peptide to be a suitable choice for exploring the aggregation kinetics and the effect of potential anti-aggregating agents by means of in vitro assays.[13,14] Interestingly, ...'

References 13 and 14 have been added to the Reference section.

2) For molecular modelling analysis human AChE  was investigated using three X-ray structures of hAChE.

Acetylcholinesterase from Electrophorus electricus was used to evaluate the inhibitory properties of the enzyme in in vitro studies. A comment is needed.

Thanks you for this remark, which has been clarified at the beginning of section 3.6 as follows:

To explore the molecular basis of the AChE inhibition, the binding mode of quercetin to AChE was investigated by combining molecular docking and MD simulations. Albeit the AChE inhibitory activity was experimentally determined using the Electrophorus electricus enzyme (eeAChE), computational studies were performed considering the X-ray crystallographic structures of hAChE deposited in the Protein Data Bank with codes 4M0F, 6CQU and 6O4X. This choice was motivated by several reasons. First, the lack of crystallographic structures for the eeAChE enzyme, since the low resolution achieved in previous studies impeded to model the details of backbone and side chains. [60] Second, the large preservation of the residues that shape the catalytic and peripheral binding sites suggest that the ligand will adopt a similar binding mode in both eeAChE and hAChE. For instance, only three differences can be noticed for the residues in the peripheral site: two correspond two conserved replacements (Ile267 and Leu289 in eeAChE by V282 and Ile294 in hAChE, respectively), and the third involves the change of Gly288 in eeAChE by Ser293 in hAChE, but this substitution is located on the edge of the binding pocket. Third, the large number of ligand complexes available with hAChE permits to identify ligand-adapted conformational changes in certain residues, such as Tyr337 in the catalytic pocket and Trp286 in the peripheral site, which can be found in distinct arrangements (Figure S1), as was noticed in earlier studies [61,62]. Keeping in mind the sensitivity of docking to the positional details of residues, it is then convenient to perform these calculations considering the X-ray structures 4M0F, 6CQU and 6O4X as templates in order to explore the effect of the conformational rearrangements of Tyr337 and Trp286 on the binding pose of quercetin.

References 60-62 have been added to the text.